# Cannulation Technique of Vascular Access in Haemodialysis and the Impact on the Arteriovenous Fistula Survival: Protocol of Systematic Review

**DOI:** 10.3390/ijerph182312554

**Published:** 2021-11-29

**Authors:** Ricardo Peralta, Luís Sousa, António Filipe Cristóvão

**Affiliations:** 1Lisbon School of Nursing, University of Lisbon, 1600-096 Lisbon, Portugal; acristovao@esel.pt; 2Comprehensive Health Research Centre, University of Évora, 7000-811 Évora, Portugal; lmms@uevora.pt

**Keywords:** buttonhole, rope-ladder, cannulation technique, arteriovenous fistula, vascular access, needling pain, systematic review

## Abstract

Background: Based on a literature review of various studies, comparisons between BH and RL are inconclusive regarding some outcomes. However, in the last 5 years, some studies have been published that may contribute to clarifying which cannulation technique (CT) allows better fistula survival. Aim: To review which cannulation technique allows better primary patency of the arteriovenous fistula in haemodialysis patients. Methods: We will include all randomised controlled trials and observational studies that include comparisons among CTs and thus define the benefits and risks of each CT. A PRISMA-compliant systematic review and meta-analysis will be performed in accordance with the quality and homogeneity of studies. A comprehensive search strategy will be applied to the CINAHL, MEDLINE and Embase electronic databases from January 2000 to September 2021. The primary outcome is the arteriovenous fistula primary patency. To assess the risk of bias in randomised controlled trials or quasi-experimental studies, we will use the tool Revised Cochrane Risk-of-Bias Tool for Randomized Trials (RoB 2). For nonrandomised studies, the Risk of Bias In Non-Randomized Studies of Interventions (ROBINS-I) will be used. Discussion: The evidence generated from this systematic review of current evidence could inform the design and implementation of continuous quality improvement programs in cannulation techniques in haemodialysis patients, as well as contributing to improving the curricula within haemodialysis courses. This protocol was registered with the National Institute for Health Research PROSPERO database prior to commencement (registration number CRD42021237050).

## 1. Introduction

A suitable vascular access (VA) is essential for the successful treatment of patients with end-stage kidney disease (ESKD) on a haemodialysis (HD) programme. With an increasing average age, depletion of vascular territory, and diabetes as the primary cause of renal aetiology, the establishment and preservation of a suitable VA has become a significant challenge. A functioning VA is the lifeline [1,2] that allows patients to undergo HD as replacement therapy for kidney function, allowing their survival and maintenance of an acceptable quality of life. Conversely, the preservation and maintenance of a complication-free VA remains the Achilles’ heel [2,3]. Moreover, vascular access dysfunctions continue to remain the major cause of comorbidities and hospitalisations [4,5,6] in ESKD patients.

The choice of cannulation technique (CT) and VA cannulation are the most important aspects in dialysis [7] and the onus is on nurses to constantly update their knowledge and skills in this area. The importance of this choice is fundamental to properly use the vascular access and allow effective treatment, and correct and appropriate arteriovenous fistula (AVF) cannulation is the key to its preservation and the prevention of VA-related dysfunction [8].

Three cannulation techniques are known: rope-ladder (RL), buttonhole (BH), and area puncture [9]. The area CT consists of repeatedly cannulating the VA in the same place, leading to decreased thickening of the vein wall and tissues, creating not only an area of greater fragility at the puncture sites but also the formation of aneurysms with increased risk of vein wall rupture [8]. This method derives from inadequate use of the RL [10]. Despite this knowledge, it is the most used technique in nine European countries, among 65.8% (44 to 77%) of patients, compared to only 28.2% for rope-ladder and 6% for BH [11].

Undoubtedly, area CT causes VA dysfunction, with consequences for its preservation, and is associated with a greater risk of VA collapse when compared to the other two methods [11]. Compared to the BH, it presents a longer haemostasis time and intensity of pain perception [12]. As mentioned before, this CT is associated with severe aneurysm development; therefore, it should be avoided whenever possible [13].

Rope-ladder has always been referenced as the preferred technique for vascular access cannulation. This technique consists of using the entire extension of the available vessel by progressive rotation of the puncture points. This method is not always used, due to resistance from patients, considering that it is very painful [14,15] and carries increased risk of bruising [16]. Furthermore, nurses also tend to avoid exploring the entire length of the vessel, for the same reasons. In addition, after a while on a dialysis programme, with the depletion of the vascular territory, in some cases, this technique is impracticable due to the limited length of the vein.

The BH technique, initially described in 1977 and later by Z.J. Twardowski [17], is referred to as a CT that is always performed in the exact same place and with the same inclination and depth. This technique has some limitations since it must be used exclusively in AVF and requires the cannulation to be performed by the same nurse until the tunnel is built and is time-consuming. Early research indicated that it was a promising CT, with reduction in the number of bruises and reduction in failed cannulation, and was preferred by nurses when compared to the other two techniques [17]. However, previous studies show that it appears to be associated with an increased risk of local infection and bacteraemia compared to the RL [18,19,20,21]. However, Chong Ren et al. [20], in a meta-analysis, failed to prove this increase due to the heterogeneity of the results. The same study also concluded that the BH technique is associated with a reduction in aneurysms, stenosis, and thrombosis. The studies are not consistent regarding the perception of pain intensity in both techniques. Some studies have reported that the participants perceived greater pain intensity with the BH than the RL, but in both studies, the patients were given a local anaesthetic [21,22]. Two systematic literature reviews and a meta-analysis were consulted [20,23,24], and the various results are inconclusive in the comparison of BH and RL, regarding the time of haemostasis, haematoma, hospitalisation, interventions, or survival of the VA. These limitations are associated with the poor quality and heterogeneity of the studies. However, in the last 5 years, some studies have been published that may contribute to clarifying which CT allows better fistula survival.

## 2. Methods and Analysis

This protocol was registered with the National Institute for Health Research PROSPERO database prior to commencement (registration number CRD42021237050).

### 2.1. Objectives

We have established the following objectives for this systematic review:

Identify which CT allows better primary patency of the arteriovenous fistula in HD patients;

Identify which CT has a lower rate of AVF complications;

Identify which CT causes less intensity of pain perception with cannulation.

To achieve the above objectives, our initial review question is:

Which cannulation technique (CT) favours better AVF survival among end-stage renal disease patients in a regular HD programme?

### 2.2. Eligibility Criteria

Our eligibility criteria will be based on the PICO framework: population—end-stage kidney disease adults under HD therapy using an AVF; intervention—cannulation in AVF by healthcare professionals or auto-cannulation; comparison—any study that compares the outcomes obtained among different CTs; outcomes—identification of outcomes related to CT, such as AVF survival, local infection, bacteraemia, thrombosis, aneurysm formation or development, haematoma, infiltration, bleeding time, haemostasis time, and pain.

### 2.3. Types of Participants/Population

The studies selected will include adults (≥18 years) with ESKD on a regular programme of HD (3 times a week) in hospitals or private clinics, with AVF as VA. We will also include patients who are being treated at home. All cannulation methods are eligible and include patients who perform auto-cannulation. For the review, we included only patients with vascular access AVF.

### 2.4. Types of Study to Be Included

We will include all randomised controlled trials (RCT) and quasi-experimental and observational studies that satisfy the following criteria:

Studies that make comparisons between CTs and thus define the benefits and risks of each CT;

Primary studies, full papers, or abstracts that refer to one or more outcomes.

### 2.5. To Be Excluded

Studies of patients with vascular access other than AVF;

Studies combining patient data from home haemodialysis with data from hospital or clinical services;

Studies with patients on acute haemodialysis;

Qualitative studies.

### 2.6. Search Methods for Identification of Studies

The following databases will be used to identify studies for the review:

In the Cumulative Index of Nursing and Allied Health Literature (CINAHL) from the Ebsco database, we will firstly use the CINHAL Headings to select the terms indexed to each descriptor, followed by the Boolean expressions AND and OR and carry out the 1st search activation. In the PubMed/MEDLINE database, the methodological procedure will be identical, using MeSH to find the terms indexed to each word and then the Boolean expressions AND and OR and carry out the 1st search activation. In the EMBASE database, we will use the same procedures as above.

Regarding the articles selected in the previous databases, but with restricted access to full texts, we will carry out another search in Google Scholar that allows us to obtain these articles or request access from the authors. We will include studies published after 1 January 2000 until the end date of the research.

### 2.7. Other Sources

We will also consider other databases, such as:

Cochrane Library, ScienceDirect web, Joanna Briggs Institute Library Evidence-Based Practice Network (JBI), SCOPUS, Researchgate, American Society of Nephrology (ASN), American Nephrology Nurses Association (ANNA), Sociedade Espanhola de Nefrologia (SEN), Sociedade Brasileira de Nefrologia (SBN).

### 2.8. Comparator Control

The outcomes associated with the cannulation techniques used in AVF in HD patients will be compared. The described and known CTs are RL, BH, and area. Previous randomised control trials used variant expressions of RL, such as traditional RL needling [25], conventional puncture technique [26], conventional different-site technique [27], usual practice of RL rotation technique [22]. In these cases, RL will be considered an undefined and different technique that is likely to be a mixture of area puncture and RL. In addition, some investigators only specified rope-ladder as the control group CT in the title of the paper and did not detail information on how the method was implemented. A new approach to AVF called the Multiple Single Cannulation Technique (MuST) has recently been published. The MuST is based on the association between the RL and BH techniques and consists of the use of the entire length of the available vessel through progressive rotation, and the use of the three specific cannulation sites for each cannulation day of the week [28,29]. We will analyse this study and the outcomes presented.

The perception of pain caused by cannulation will not be compared between CTs in patients who use local anaesthetic creams unless they use the same products and protocols.

To standardise the reports, the event variation will be reported according to the frequency of the event for 1000 days/AVF, which is equivalent to 1000 patients per day. For the characterisation of the benefits and risks of each technique, it is necessary to carry out assessments of indicator answers/outcomes through which the groups are compared.

### 2.9. Context

The systematic review is focused on CTs performed by healthcare professionals on patients with AVF undergoing HD. Studies conducted in hospitals, in private clinics, and on home haemodialysis patients will be included.

### 2.10. Primary Outcomes

The primary outcome is the AVF primary patency and this will be measured by the percentage of fistulas in use from the beginning of the study to the date of the first clinical intervention for angioplasty or vascular surgery (unassisted patency).

Referral for endovascular intervention was based on one or more indicators of AVF dysfunction:

Changes in physical examination;

Qa < 400 mL/min;

Increased haemostasis time (>10 min);

Cannulation failure: failure or inability to insert the dialysis needles;

Decreased dialysis efficacy: deficient dose of HD, spKt/V < 1.2.

Referral for surgical intervention was based on one or more indicators of AVF dysfunction:

Rupture of the AVF wall or thrombosis;

Progressive development of aneurysm;

Acute bleeding;

Local AVF infection;

Deficient distal perfusion with signs of ischaemia.

### 2.11. Secondary Outcomes

As secondary outcomes, we consider the following:

AVF survival, for which failure was defined as an AVF no longer used for successful HD (also known as assisted patency);

Local AVF infection and bacteraemia (local infection defined by flushing, oedema, or local exudate; bacteraemia shown by positive laboratory results);

Haematoma or infiltration (an incident occurring during cannulation with local oedema, pain, or increase in venous pressure and requiring further cannulation);

Haemostasis time (defined as time until bleeding stopped after needle removal at the end of treatment);

Development of aneurysms (dilatation of a vein segment to 3 times the diameter of the segment considered normal, which means a segment with width ≥1.8 mm [30]);

Stenosis (narrowing of the vessel lumen);

Prolonged bleeding during HD;

Any other adverse outcomes related to CT and reported;

Local pain related to CT (assessed after cannulation according to visual analogue scale or other scale or results from qualitative analysis).

### 2.12. Data Extraction (Selection and Coding)

The evaluation of the titles and the abstracts during the selection of the studies will be carried out independently by a single reviewer (RP), with subsequent verification by a second reviewer (AC). All duplicate studies will be deleted immediately. This selection will be made strictly according to the inclusion and exclusion criteria defined in this protocol. When the title and abstract are not sufficiently enlightening, a new search will be carried out looking for the entire article or requesting it from the authors, thus reducing the risk of not selecting important articles. After this selection, the full versions of the potentially eligible articles will be extracted. Any disagreement or inconsistencies will be resolved by discussion between the two authors and, if not resolved, opinion will be sought from the third reviewer. In the case of a lack of important data, they will be requested from the authors of the study. If, after several attempts to contact the authors, the missing data cannot be obtained, we will analyse only the available data and discuss the impact on the final outcomes (Figure 1). The screening process and results will be presented in a Preferred Reporting Items for Systematic Reviews and Meta-Analysis (PRISMA) flow chart [31].

### 2.13. Risk of Bias Assessment

To assess the risk of bias in randomised controlled trials or quasi-experimental studies, we will use the tool Revised Cochrane Risk-of-Bias Tool for Randomized Trials (RoB 2) [32]. Based on the results of the evaluation scale, studies with low methodological quality will be excluded depending on the number of articles selected and the evidence found. Therefore, each study will be categorised as low risk, high risk, or unclear risk of bias. For nonrandomised studies, the Risk of Bias In Non-Randomized Studies of Interventions (ROBINS-I) [33] will be used.

### 2.14. Strategy for Data Synthesis

The articles selected during the systematic review will be presented in a summary table with the following main characteristics: name of authors, year of publication, country, characteristics of study, sample size (n), and outcome analysis method.

Outcomes: characteristics of participants (mean age, comorbidities), type of dialysis (hospital, clinic or home), duration of the study (months), and primary and secondary outcomes obtained including the methodology used for pain perception.

We will perform a meta-analysis only if the included studies are sufficiently homogeneous in terms of participants, interventions, and outcomes. When statistical comparison is not possible, the results will be presented in a narrative form, including tables and figures to assist in the presentation of the data, when appropriate.

Meta-analysis will be performed using the generic inverse variance method using Cochrane Collaboration Review Manager software (RevMan).

### 2.15. Analysis of Subgroups or Subsets

Due to the inclusion of RCTs and observational studies, we will carry out the treatment of the data separately. In the observational studies, we will summarise the events in each cannulation technique group and compare the frequency and the interquartile range of the events. When appropriate, we will use the tests described above.

## 3. Discussion

The results of previous systematic reviews are not clarifying due to the heterogeneity of the RCTs, providing disparate results. Our systematic literature review protocol intends to research new studies that have since been published, comparing different CTs and contributing to the clarification of safe cannulation practices that allow for better AVF survival. The findings of this review may help to conduct future research, identifying methodological limitations and knowledge gaps in the literature available to date. Future studies can be designed to minimise limitations and gaps and help to improve clinical practice.

## 4. Conclusions

Our protocol of systematic review intends to lay out the method planned for reviewing the information on the benefits and risks associated with each AVF cannulation technique in HD patients. It is a feasible mean to synthesize the broad evidence available on the topic, and the systematic review allows interpreting results of RCT and observational studies within the evidence. By summarizing all related studies, it improves understanding of inconsistencies of evidence. Apart from identifying the research gaps, the review will help to provide evidence based for knowledge translation so that the result will be used for clinical practice and HD course curricula.

## Figures and Tables

**Figure 1 ijerph-18-12554-f001:**
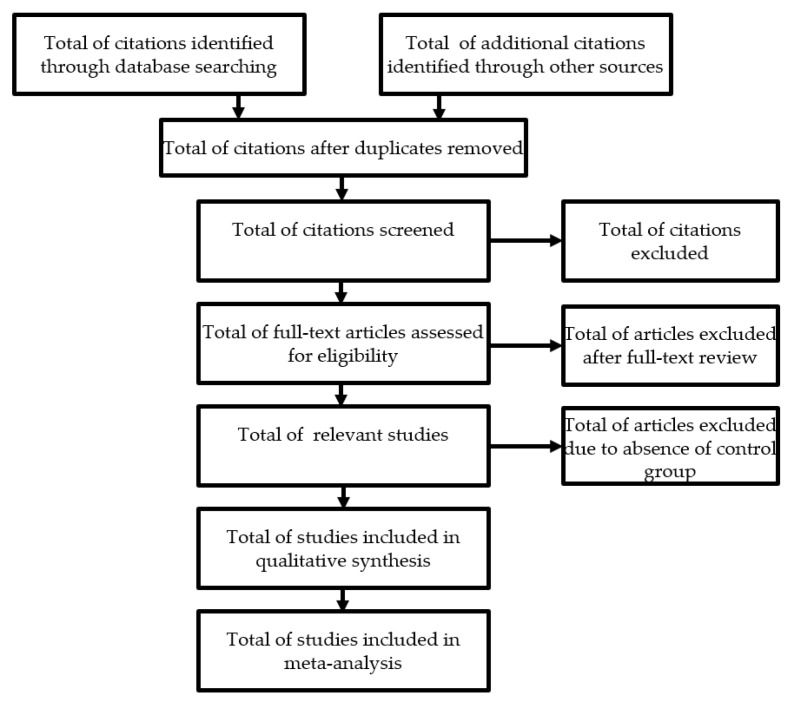
Flow diagram documenting inclusion and exclusion of studies.

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
