# Peer review of "Cannulation Technique of Vascular Access in Haemodialysis and the Impact on the Arteriovenous Fistula Survival: Protocol of Systematic Review"

_ijerph, 2021, doi:10.3390/ijerph182312554_

Round 1
Reviewer 1 Report
The paper written by the following Authors: Ricardo Peralta, Luís Manuel Mota de Sousa, António Filipe Cristóvão, entitled “Cannulation technique of vascular access in haemodialysis and the impact on the arteriovenous fistula survival: Protocol of systematic review” presents an interesting study on which cannulation technique allows a better primary patency of the arterio venous fistula in hemodialysis patients.
Although the paper is interesting, I have some major concerns:
Title
The title reflects the results presented here.
Abstract
The abstract is lacking informative conclusion. It should be written in more details.
Results and Discussion
There are no results and discussion presented in the manuscript. It should be included in the article. Currently there is only description of planned activities.
Conclusions
Conclusions should be written in more details.
Author Response
Response to Reviewer 1 Comments:
The paper written by the following Authors: Ricardo Peralta, Luís Manuel Mota de Sousa, António Filipe Cristóvão, entitled “Cannulation technique of vascular access in haemodialysis and the impact on the arteriovenous fistula survival: Protocol of systematic review” presents an interesting study on which cannulation technique allows a better primary patency of the arteriovenous fistula in hemodialysis patients.
Pont 1: Although the paper is interesting, I have some major concerns:
Title
The title reflects the results presented here.
Abstract
The abstract is lacking informative conclusion. It should be written in more details.
Response 1: We thank the reviewer for this suggestion. On pag. 1 line 22, it was inserted: “Discussion: The evidence generated from the systematic review of current evidence could inform the design and implementation of continuous quality improvement programs in cannulation technique in hemodialysis patients. As well as contributing to improve curricula within hemodialysis courses.”
Pont 2: Results and Discussion
There are no results and discussion presented in the manuscript. It should be included in the article. Currently there is only description of planned activities.
Response 2: We thank the reviewer for this commentary, but this manuscript is a protocol for a systematic review and, therefore, there are no results to present. However, we hope to find some randomized controlled trails and observational studies that provide us important results and may clarify the research question.
Pont 3: Conclusions
Conclusions should be written in more details.
Response 3: We thank the reviewer for this suggestion. On pag. 6 line 272 it was inserted:
“The findings of this review may help to conduct future research, identifying methodological limitations and knowledge gaps in the literature available to date. Future studies can be designed to minimize limitations and gaps and help improve clinical practice and hemodialysis course curricula.”
Reviewer 2 Report
Dear Editor,
Thank you for invitation to review the manuscript by Ricardo Peralta et al. In summary it is a description of statistical tools and methods which will be used to analyse a number of available studies aiming on native AVF or prosthetic AVG needling technique. It is very important problem in renal replacement therapy as the procedure influences a vascular access survival. However the lack of results of the analysis it the major manuscript weakness and thus makes it ineligible for publication.
Author Response
Response to Reviewer 2 Comments:
Thank you for invitation to review the manuscript by Ricardo Peralta et al. In summary it is a description of statistical tools and methods which will be used to analyse a number of available studies aiming on native AVF or prosthetic AVG needling technique. It is very important problem in renal replacement therapy as the procedure influences a vascular access survival.
Pont 1: However the lack of results of the analysis it the major manuscript weakness and thus makes it ineligible for publication.
Response 1: We thank the reviewer for this commentary, but this manuscript is a protocol for a systematic review and, therefore, there are no results to present. However, we hope to find some randomized controlled trails and observational studies that provide us with important results and may clarify the research question.
Reviewer 3 Report
This is a nice paper. However, I have some comments.
The findings from this paper are excellent and worthy to review.
This manuscript contained some questions described below.
I think this paper is interesting, this review contributes to future's clinical
medicine largely. I have some questions from a point of view of clinical medicine.
It is being examined based on many documents. However, it is a difficult context to understand because there are many sentences. Please give a clear explanation using charts. After all, does it mean that there is no significant difference between RL and BH? Do you think that the puncture method should be selected according to the individual case in the end?
Author Response
Response to Reviewer 3 Comments:
This is a nice paper. However, I have some comments. The findings from this paper are excellent and worthy to review. This manuscript contained some questions described below. I think this paper is interesting, this review contributes to future's clinical medicine largely. I have some questions from a point of view of clinical medicine.
It is being examined based on many documents. However, it is a difficult context to understand because there are many sentences.
Pont 1: Please give a clear explanation using charts.
Response 1: We thank the reviewer for this suggestion that improved the manuscript. On pag. 5 line 234 it was inserted:
Figure 1. Flow diagram documenting inclusion and exclusion of studies.
Pont 2: After all, does it mean that there is no significant difference between RL and BH?
Response 2: We thank the reviewer for this question. Compared with RL, BH can significantly reduce the formation of aneurysm, thrombosis and stenosis, but there were no pain or intervention reduction for fistula prevention in patients with arteriovenous fistula (AVF). The efficacy of BH in infection and hematoma control, bleeding time reduction and fistula survival rate, needs further study. However, a very recent RCT showed a greater survival AVF using RL. Other recent studies recommend the exclusive use of BH in patients with short fistula courses or self-cannulation due to increased infection.
Pont 3: Do you think that the puncture method should be selected according to the individual case in the end?
Response 3: We thank the reviewer for this question that we really appreciated it. It is an excellent and interesting question, we have been defending this in clinical practice. Each patient has different fistula paths and the cannulation technique (CT) must be selected according to the limitations or dysfunctions of the vein. However, area CT should always be avoided. A new CT that appears to be promising, called multiple single cannulation technique (MuST), was recently published.
Available in: https://doi.org/10.1111/hdi.12962

Round 2
Reviewer 1 Report
I accept the manuscript in the present form.
Reviewer 2 Report
Dear Editor,
I have read the revised version of manuscript by Ricardo Peralta et all. The paper has been improved substantially. I accept all answers and revision done by authors, therefore I feel its present form is worth of publication. I look forward the results.
Reviewer 3 Report
This paper is well written and informative
I have no specific comments.